# Explaining the Use of Social Network Sites as Seen by Older Adults: The Enjoyment Component of a Hedonic Information System

**DOI:** 10.3390/ijerph16101673

**Published:** 2019-05-14

**Authors:** Patricio Ramírez-Correa, Elizabeth E. Grandón, Muriel Ramírez-Santana, Leonard Belmar Órdenes

**Affiliations:** 1Engineering School, Universidad Católica del Norte, Coquimbo 1780000, Chile; patricio.ramirez@ucn.cl; 2Department of Information Systems, Universidad del Bío-Bío, Concepción 4030000, Chile; egrandon@ubiobio.cl (E.E.G.); leonardbelmar@gmail.com (L.B.Ó.); 3Department of Public Health, Faculty of Medicine, Universidad Católica del Norte, Coquimbo 1780000, Chile

**Keywords:** older adults, SNS, hedonic information systems, Chile

## Abstract

Previous studies suggest that older adults are living increasingly alone and without the company of their close relatives, which cause them depression problems and a detriment to their health and general wellbeing. The use of social network sites (SNS) allows them to reduce their isolation, improve their social participation, and increase their autonomy. Although the adoption of various information technologies by older adults has been studied, some assumptions still predominate, for example, that older adults use SNS only for utilitarian purposes. However, considering SNS as hedonic information systems, and in order to extend the theoretical explanation of the intention to use hedonic systems to their actual use, this study aims to determine the influence of perceived enjoyment, perceived usefulness, and perceived ease of use on the use of SNS by elders in Concepción, Chile. Two hundred fifty-three older adults participated in the cross-sectional study. The results indicate that perceived ease of use is the variable that has the greatest total effect in explaining the use of SNS and that by adding the perceived enjoyment construct, the explanatory power of the model increases significantly. Therefore, advancement in user acceptance models, especially in the use of SNS by elders, can be made by focusing on the type of system, hedonic or utilitarian.

## 1. Introduction

Since its creation, social network sites (SNS) have spread to an extraordinary speed [1], becoming established as a widespread and permanent phenomenon in our society [2]. In its beginnings, SNS were associated to user´s personal life [3], however, today its influence is of a very wide spectrum, including processes of political communication [4] with the distribution of financial information of large corporations [5]. SNS are defined as web-based services that allow people to build a public or semi-public user profile within a closed system, with the purpose of managing a list of users with whom they share a connection through creating and consuming content [6]. By establishing social relations over the network, SNS users allow the emergence of a different type of social structure [7], which in one way or another influence its members. In fact, the decision to use SNS depends to a large extent on the interactions between the users [8]. The fact that SNS are based on technology [9], its popularity [10], and its unique characteristics [11], make them a topic of particular interest.

A particular case of significant relevance is the adoption of SNS by older adults. According to the United Nations, the proportion of the world population of more than 60 years increases in accelerated form and is expected to be doubled by the year 2050, from 12.7% in 2017 to 21.3% in 2050 [12]. In particular, Chileans that surpass 60 years are 16.2% of the total population [13], reflecting a strong transformation of the population pyramid of the country, from a triangular form in 1975 to a rectangular form in our days [14]. The Chilean ageing index indicates that there is a ratio of 86 people over 60 years for every 100 under 15 years [15], a value clearly higher than the average of Latin America (39.6), and close to the other countries of the organization for economic cooperation and development [16]. In this context, studies point out that older adults are living more and without the company of their close relatives [12]. This causes problems of depression and, therefore, a detriment to the health and general wellbeing of this age group. For this reason, a series of digital technologies have been developed in order to obtain health benefits and improve the communication between patient and health care services (eHealth) [17]. Some of these tools have focused on improving safety, optimizing the medical management of some pathologies [18,19], and supporting older patients to improve the self-care and self-monitoring of their pathologies [20]. Multiple applications allow, for example, to warn in real time when an older person suffers a fall; monitor vital signs, blood glucose and oxygen levels; send reminders for medical appointments or time to get vaccinated [18,21,22]. In particular in the case of SNS, the medical literature emphasizes its effectiveness in achieving improvements in the quality of life of older adults [23]. Previous research has shown that the use of SNS by older adults allows them to reduce isolation, improve social participation, and increase autonomy [24,25,26]. From a health standpoint, older adults who make use of SNS tend to have fewer chronic pathologies and fewer depressive symptoms; while increasing their perception of wellbeing [25,26]. For example, older Slovenians relate the reduction of their loneliness levels to their participation in SNS [27], and older Chinese feel satisfied with life since the use of SNS allows them to reduce isolation and increase their autonomy. [25].

Considering the positive effects that SNS can have on the wellbeing of older adults [22,23,24], this study intends to determine the factors that influence its use. To this end, and considering that SNS have an enjoyment component, this research focuses on the hedonic side of this technology and proposes a research model based on an extension of the popular technology acceptance model (TAM) [28]. The rest of the paper is organized as follows: Section 2 presents previous studies and proposes the research model and hypotheses. Section 3 and Section 4 shows the research methodology and the results obtained. Finally, discussion, conclusions, and limitations of the study are offered.

## 2. Literature Review

General adoption models such as TAM [28] has been frequently used to study the use of information technologies by older adults. TAM highlights the importance of two individual perceptions, usefulness and ease of use, as antecedents of intention and subsequent use of technology [29,30,31,32]. According to the TAM theory, perceived usefulness is defined as the degree to which a person believes that using a particular system would enhance his or her job performance. Perceived ease of use, on the other hand, is defined as the degree to which a person believes that using a particular system would be free of effort [28]. TAM is a robust model that has been found to explain the adoption of technology in different domains. Davis [26] found that perceived usefulness and perceived ease of use highly correlate with future usage of information technology (*r* = 0.85 and *r* = 0.59 respectively). The high heterogeneity in older adults [18,33] has generated contradictory results in the application of traditional adoption models [31]. This fact has led to use psychological criteria, such as cognitive age, technological anxiety, or level of audacity, to better explain the differences among older people when using information technology [33]. In this line of thought, it is possible that using a specific conceptualization for the adoption of SNS in adults enables us to improve the explanation of the phenomenon [26]. Even though the adoption of various information technologies by older adults have been studied, stereotyped assumptions associated with old age still predominate [34]. For instance, that SNS are used only by young people and older people use them only for utilitarian purposes. In this academic context, and according to various authors, SNS can be conceptualized as hedonic information systems [35,36,37]. A hedonic information system is intended to deliver a self-realizable value to its end users, and in its purest form, interaction with these systems is designed to be an end in itself [38]. As indicated by Van der Heijden [38], the objective of hedonic systems is to provide self-fulfilling value to the user, as opposed to utilitarian systems, of which the purpose is to provide instrumental value to the user. In today’s society, many information technologies play a hedonic role [36,37,38,39], which has driven the scientific interest in this type of systems [40,41,42] and that its acceptance is established as a relevant study space [40]. In general, the degree to which the user experiences fun when using a hedonic system determines its value, and therefore, if the user experiences more fun using it in the present, he or she will have greater intention of continuing the use of the system in the future.

In his seminal work on the acceptance of hedonic information systems, Van der Heijden points to the need to understand and explain the reasons for their use by consumers around the world [38]. According to this author, empirical evidence suggests that information systems are easier to use when they are visually more attractive. Van der Heijden [38] studied the acceptance of a movie website conceptualized as a hedonic information system based on TAM, and its results indicate that both perceived ease of use and perceived enjoyment are the most important antecedents to explain the intention to use this type of systems [38]. Perceived enjoyment is defined as the extent to which the activity of using the computer is perceived to be enjoyable in its own right, apart from any performance consequences that may be anticipated [43].

Other authors have studied the acceptance of SNS conceptualized as hedonic information systems. For example, Rosen and Sherman [35] analyzed how the use of SNS can be explained, considering that its use hinders personal productivity, which is inconsistent with one of the antecedents, perceived usefulness, that determine the intention of using technologies according to the TAM theory. Their results indicate that both perceived enjoyment and perceived ease of use are the antecedents of the use of SNS. On the other hand, Lin and Lu [37] studied the intention to continue using SNS and their findings suggest that together with the opinions of friends, enjoyment is the most important factor in determining the intention to continue using SNS. Finally, Xu and colleagues [44], in a sample of university students and through the analysis of various determinants of the use of SNS, found that user utilitarian gratifications of immediate access and coordination, and hedonic gratifications of affection and leisure were positive predictors of SNS usage [36].

These results are aligned with other studies that have focused on older adults. It has been reported the existence of social motivations, utilitarian, and hedonic variables that influence the use of SNS [45]. In Chile, for example, a recent study carried out from secondary data of a representative universe of SNS users, estimated that around 40% of older adults have hedonic motivations for social networks sites use [46]. Considering the findings of the aforementioned research, and in order to extend the theoretical explanation of the intention to use hedonic systems to the current use of them, this study aims to determine the influence of perceived usefulness, perceived ease of use, and perceived enjoyment on the use of SNS by older adults in Concepción, Chile. As a theoretical basis, ithas been taken the TAM model [24] in conjunction with the proposal of Van der Heijden [36] and studies of Ernst [47], and we have formulated the following hypotheses:
**H1:** Perceived ease of use is positively related to perceived usefulness of SNS by older adults in Chile.
**H2:** Perceived ease of use is positively related to the use of SNS by older adults in Chile.
**H3:** Perceived ease of use is positively related to perceived enjoyment of SNS by older adults in Chile.
**H4:** Perceived usefulness is positively related to the use of SNS by older adults in Chile.
**H5:** Perceived enjoyment is positively related to the use of SNS by older adults in Chile.

Figure 1 shows the research model with the associated hypotheses.

## 3. Materials and Methods

The research was carried out in two stages: First, a pilot study was conducted in order to validate the measurement scale of the instrument utilized for data collection. The scales were adapted from the study of Davis [24] to measure perceived usefulness, perceived ease of use, and actual use, and from the study of Davis et al. [43] to measure perceived enjoyment. Once the scale was validated, a cross-sectional study was carried out in September 2018. Convenience sampling technique was used to collect data of 253 elders. As in the pilot study, the survey was applied through personal interviews where older adults came from different points of Concepción city, particularly from training and social older adults’ centers of the community. Before starting the survey, and as in the pilot study, older adults were consulted about their age range and if they were currently using social networks. Following References [48,49], older adults were considered people from 50 years and above. Anonymity of the respondents was guarantee in the data collection process.

The instrument consisted of two parts: A demographic section where age, gender, marital status, work condition (retired or not retired), education level, and years of experience using SNS were asked. A definition of SNS was included in the survey along with a question regarding the SNS utilized (Facebook, Instagram, Twitter, WhatsApp, other). This section also included different uses of SNS such as to chat, play, watch videos, and so on. The second section consisted of questions to measure the main constructs of the study: Perceived ease of use, perceived usefulness, perceived enjoyment, and actual use of SNS. A 7-point Likert scale was used with answers ranging from 1: “strongly disagree” to 7 “strongly agree”. Table 1 below shows all the items used to measure the four constructs of the study.

As mentioned previously, the high heterogeneity among older adults has generated contradictory results in the application of technology acceptance models. In this line of ideas, we proposed a research model based on the conceptualization of an SNS as a hedonic information system with the purpose of improving the explanation of the use of this technology in this segment of users. For the above reason, we analyze this phenomenon by examining the hypotheses in a causal model of latent variables. Structural equation modeling (SEM) was used as a tool for statistical analyses. In particular, partial least square (PLS) regression technique was utilized using SmartPLS 3.2 software [50] to test the proposed research model. The PLS technique is chosen for different reasons: (1) PLS has been previously used in research related both hedonic information systems [40] and acceptance of information technologies by older adults [29]; (2) the use of PLS has been recommended for causal applications when theoretical knowledge about a subject is scarce [50]; (3) PLS can estimate models with small samples, as is the case of older adults studies. Two models define PLS: The measurement model and the structural model. 

The first one analyzes the reliability and validity of the instrument and the second one analyzes the relationships between the constructs or latent variables. The following section shows the results associated with these two models for the cross-sectional study.

### Ethics Statement

Participation in the study was voluntary. All study participants were informed about the anonymity and confidentiality of their responses. According to standard socio-economic studies, no ethical concerns are involved other than preserving the anonymity of participants. This procedure was approved by both the head of the School of Engineering of the Catholic University of the North and the head of the Department of Information Systems of the University of Bío-Bío. The study was carried out guaranteeing the protection of the ethical principles in the research declared by the Declaration of Helsinki of the World Medical Society, 1964 and subsequent revisions; as well as the principles declared in the Universal Declaration on Bioethics and Human Rights, UNESCO, 2005.

## 4. Results

The results of the pilot and cross-sectional studies are shown below.

### 4.1. Results from the Pilot Study

A total of 47 face-to-face interviews were conducted to elders in Concepción, Chile. In order to meet the requirement criteria to be a subject of the study, elders were first asked their age range and if they currently use SNS. Out of 47 surveys, one was eliminated due to incomplete answers. Fifty percent of the respondents were male with an average age of 56.8 years old. Most of the respondents had tertiary education level (48%) with an average of seven years of experience using SNS.

We corroborated the reliability and validity of the instrument based on the model TAM. The constructs were evaluated using Cronbach’s Alpha and composite reliability indicators. The average variance extracted (AVE) indicator was utilized to evaluate the convergent validity of the instrument. All the indicators obtained from the pilot test were satisfactory according to the criteria defined by [51]. Therefore, reliability and convergent validity of the scale is met in the pilot sample.

### 4.2. Results from the Cross-Sectional Study

A total of 253 interviews were completed for the cross-sectional study. The majority of completed surveys were of women (69.2%) and had an average age of 60.1 years old. A relative majority of the respondents (64.4%) stated that they had a tertiary education level, and 5.8 years of experience using SNS. See Table 2 for more detail. Table 3 indicates the assessment of the measurement model.

Table 3 indicates the assessment of the measurement model. Three indicators are shown in the table: (1) Cronbach´s Alpha, a measure of internal consistency reliability that assumes equal indicator loadings, where values above 0.7 are acceptable; (2) composite reliability, a measure of internal consistency reliability that does not assume equal indicator loadings; in its place, it takes indicator loadings into consideration in its calculation, and values greater than 0.7 are acceptable; and (3) average variance extracted (AVE), which is a measure of convergent validity, defined as the degree to which a construct explains the variance of its indicators, and values above 0.5 are acceptable.

The results, with respect to the analysis of research model and the relationships proposed, are indicated in Table 4 and Figure 2. The R^2^ indicates the amount of variance of the dependent variables that is explained by the variables that predict it. The β coefficients indicate the extent to which the independent variables contribute to the explained variance of the dependent variables. The significance of the β coefficients was calculated using bootstrapping. Bootstrapping is a technique used to determine standard errors of coefficient estimates to evaluate the coefficient’s statistical significance without relying on distributional assumptions.

In relation to older adult users of SNS, the results of the PLS analysis support all the hypotheses of the proposed research model. The strongest relationship found was between perceived ease of use and perceived usefulness, and it is the perceived ease of use that has the greatest effect on the use of SNS. The analysis indicates that the perceived ease of use, perceived usefulness, and perceived enjoyment explain 42% of the SNS use by elders.

Additionally, following the procedure indicated in Reference [51], we evaluate the mediator effect of both perceived usefulness and perceived enjoyment in the relationship between perceived ease of use and SNS use. For this purpose, the size of the indirect effect in relation to the total effect was estimated through the assessment of the variance accounted for (VAF). VAF values below 20% indicate that the direct effect is very strong and there is no mediation, VAF values between 20% and 80% indicate partial mediation, and VAF values above 80% indicate full mediation. The results indicate VAF values of 9% and 21% for the variables perceived enjoyment and perceived usefulness, respectively. Consequently, there is a partial mediation of perceived usefulness between perceived ease of use and SNS use.

## 5. Discussion

The results of the study indicate that the use of SNS, conceived as hedonic system, can be predicted by the antecedents of the model TAM, perceived ease of use and perceived usefulness, and by the perception of enjoyment of using the system. All the hypothesized relationships set in the study were confirmed. As in the TAM, it was found that perceived ease of use is positively related to perceived usefulness of SNS by elders in Chile (β = 0.60, *p* < 0.001) (H1). Congruent with the results obtained by Ramirez et al. [52], who utilized the TAM to determine the use of SNS in a sample of young Chileans, this study found that perceived usefulness is positively related to the use of SNS (β = 0.27, *p* < 0.05) (H4), however, different from the same study, perceived ease of use was found to be a strong predictor of SNS use (β = 0.37, *p* < 0.001) (H2). The reason for this difference may be due to the sample age. Ramirez et al. [48] surveyed young people from generation Z, who may be familiar with SNS technology, and therefore, they use it regardless of how they perceive it. Elders, on the other hand, based their decision to adopt SNS on how easy or difficult they perceived SNS. The greatest their perception, the stronger their actual use of SNS. In fact, a recent study reports that age is an important factor influencing the use of eHealth systems [20], probably due to the lower skills of older adults to manage technology, along with variables such as low education, low income, and living in rural areas, a situation of interest in developing countries.

An interesting finding of this study validates the idea that perceived ease of use plays a critical role in the use of hedonic information systems such as SNS [35]. The construct helps perceived usefulness in contributing utilitarian value, and it helps perceived enjoyment in contributing hedonic value (β = 0.37, *p* < 0.001) (H3). Finally, and as pointed out by Ernst [48] and Azahrani et al. [53], perceived enjoyment is positively related to the actual use of hedonic systems by older people (β = 0.17, *p* < 0.01) (H5).

This study fills the lack of research concerning SNS use by elders, particularly in Chile, and complements the study of Ramirez-Correa et al. [52] who investigated the factors that influence the adoption of SNS by Chilean generation Y. By adding the perceived enjoyment construct, the explanatory power of the model increases from 30% to 42%. Therefore, advancement in user acceptance models, especially in the use of SNS by elders, can be made by focusing on the nature of system use (utilitarian or hedonic), besides the inclusion of perceived enjoyment construct. Knowing the characteristics that influence the use of technologies in older adults will allow for the adaptation of the applications to favor their use in this population. Specifically, in health, it is possible to support traditional treatment and follow-up with the use of technologies so that the patient himself takes charge of his health [20]. This is known as self-management and includes, among other actions, self-monitoring of blood pressure, glycaemia, oxygen levels, or other parameters, in patients with chronic conditions [54]. It has been shown that the use of these platforms favors the health outcomes of the people who use them [55]; however, it requires initial and permanent support from trained health personnel [17,54]. The health professional should then consider the leverage effect that an enjoyable and useful SNS application would have in an older adult, in a way that the patient will be encouraged to better engage with his or her self-management of a certain health condition.

## 6. Conclusions

In conclusion, this study found that the variables that increase the use of SNS by older persons are, in order of importance: Perceived ease of use, perceived enjoyment, and perceived usefulness. Perceived ease of use stands out as the variable that has the greatest total effect in explaining the use of SNS. It affects the use directly and indirectly through perceived enjoyment and perceived usefulness. These findings suggest that the analysis using theoretical models adjusted to the phenomenon of adoption in older adults allows enhancing the understanding of this reality. It also indicates that improvements in the perceptions of ease of use of SNS significantly affect its use for this population segment. Positive results regarding perceived usefulness are of great interest when developing eHealth systems that would contribute to improve the health situation of older adults.

In addition to being a cross-sectional study, an important limitation is the sample is that it is biased toward people with a tertiary level of education and women. Future studies on the use of SNS by older adults could consider other socioeconomic variables, together with the analysis of heterogeneity within this segment. One possible way is to use latent class segmentation techniques to determine different types of users (such as Finite Mixture Partial Least Squares (FIMIX-PLS) or Partial Least Squares Prediction Oriented Segmentation (PLS-POS)). In addition, replications of this study in culturally different countries are needed to confirm these empirical findings. Finally, future studies could include in the analysis the relationship between the adoption of SNS and the network of people with whom the older adult communicates habitually in the real world. These studies could be an interesting way to broaden the vision of the phenomenon.

## Figures and Tables

**Figure 1 ijerph-16-01673-f001:**
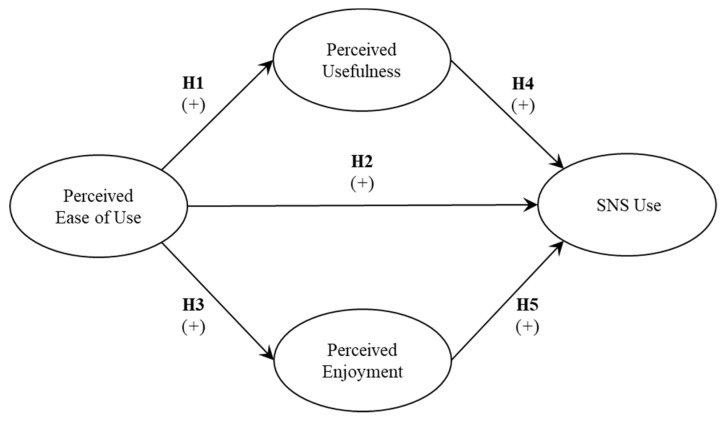
Research model.

**Figure 2 ijerph-16-01673-f002:**
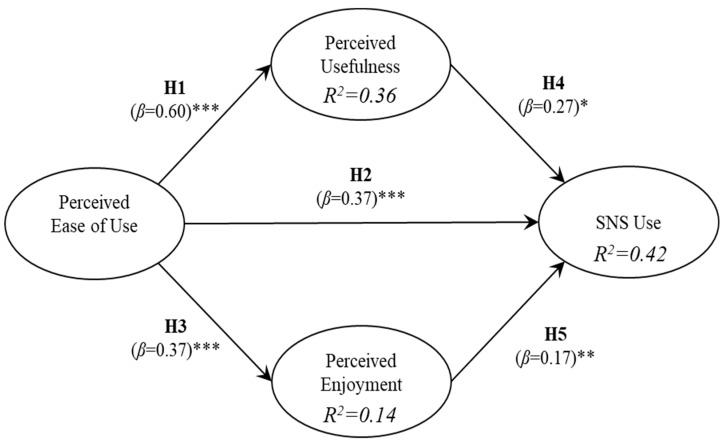
Model for older adults—users of social network systems (SNS). Statistical significance: * *p* < 0.05; ** *p* < 0.01; *** *p* < 0.001.

**Table 1 ijerph-16-01673-t001:** Items used to measure the constructs.

Variable	Item	Description
Perceived Ease of Use	PEOU1	Learning to use social networks is easy for me
	PEOU2	The process of using social networks is clear and understandable to me
	PEOU3	I find social networks easy to use
	PEOU4	Using social networks does not require much effort
Perceived Usefulness	PU1	The use of social networks allows me to access more information and people
	PU2	The use of social networks improves my efficiency in the exchange of information and in relating to others
	PU3	Social Networks are a useful service for my communication
	PU4	Social Networks are a useful service for my interaction with other members
Perceived enjoyment	PE1	The use of social networks is fun
	PE2	The use of social networks is pleasant
	PE3	The use of social networks is very entertaining
Use	USE1	I tend to use social networks frequently
	USE2	I spend a lot of time in social networks
	USE3	I get involved a lot in social networks

**Table 2 ijerph-16-01673-t002:** Distribution of the variables of interest in older people. Concepción, Chile, 2018.

Variable		*N*	%
Gender			
	Male	78	30.8
	Female	175	69.2
	Total	253	100
Education level			
	Primary	16	6.4
	High School	74	29.3
	Tertiary	163	64.4
	Total	253	100
Age		Mean 60.1 ± 9.50	
		Range 50–88 years	
Experience using SNS		Mean 5.8 ± 3.02	
		Range 1–20 years	

**Table 3 ijerph-16-01673-t003:** Validity and reliability of the cross-sectional study.

Variables	Cronbach’s Alpha	Composite Reliability	AVE
Perceived ease of use	0.914	0.939	0.795
Perceived usefulness	0.916	0.941	0.801
Perceived Enjoyment	0.946	0.965	0.902
Use	0.815	0.890	0.731

**Table 4 ijerph-16-01673-t004:** Relationship by β coefficients.

Relationship	β Coefficient (Sig.)
H1: Perceived Ease of Use ➔ Perceived Usefulness	0.60 (***)
H2: Perceived Ease of Use ➔ SNS Use	0.37 (***)
H3: Perceived Ease of Use ➔ Perceived Enjoyment	0.37 (***)
H4: Perceived Usefulness ➔ SNS Use	0.27 (*)
H5: Perceived Enjoyment ➔ SNS Use	0.17 (**)

Statistical significance: * *p* < 0.05; ** *p* < 0.01; *** *p* < 0.001.

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
