# Peer review of "Explaining the Use of Social Network Sites as Seen by Older Adults: The Enjoyment Component of a Hedonic Information System"

_ijerph, 2019, doi:10.3390/ijerph16101673_

Round 1

Reviewer 1 Report

I enjoyed reading the manuscript. Below, I would like to suggest a few things that might be helpful for revising the paper.

1.      Title: I believe the title does not need to include the location and the year of research setting. Hence, I may delete ‘Concepcion, 2018’ from the title.

2.      Pg 1, line 30: It is unclear what the authors mean by ‘limits of the personal life of its users’. Does it mean that SNS contents were mostly limited to users’ personal lives? Please explain.

a.       line 32: large corporations

b.      line 36: What does ‘a single type of social structure’ mean? Please explain the type that emerges among the SNS users.

3.      Pg 2, line 63-71: The sentences contain grammatical mistakes.

a.       line 78: the adoption of SNS in older adults

b.      line 80-82: The sentence is grammatically problematic.

c.       line 94-95: ‘That’ appears three times in one sentence. Need to be rephrased.

4.      Pg 3, line 107-110: The sentence is too long and unclear. Please rephrase.

a.       line 113: carried out from

b.      line 114: need period, instead of comma.

c.       Line 115: How do the authors extend the theoretical explanation in the current research? Be specific.

d.      Line 115-120: Unclear sentences and hard to understand. Please rephrase.

e.       The variable names need to be consistent across the manuscript. The variable names in Figure 1 are completely different from those in the hypotheses. The variable names in Table 1 and Table 3 are inconsistent as well.

f.       As the authors suggest the mediating effects of perceived usefulness and perceived enjoyment, the mediating relationships need to be hypothesized.

5.      Pg 5, Table 3: Please explain more about the findings in Table 3. How should the composite reliability and AVE indicators be interpreted? How can we determine the reliability and validity of data using the indicators?

a.       Line 188: The authors show three R-squared values in Figure 2, and it is not clear which one is related to which relationships. Please specify.

b.      line 191: Please explain the Bootstrapping method briefly and include citation.

6.      Pg 6, line 200: Since the authors suggest the mediating effect of perceived usefulness and perceived enjoyment, these effects need to be hypothesized and tested accordingly. Please use Sobel test or other appropriate methods to analyze the mediating effects.

a.       line 202: indicate that the use of

7.  Pg 7, line 229: The authors elaborated on the theoretical implications of the research, but do not seem to mention practical implications much. Given the findings, what should practitioners consider when trying to encourage elders in Chile to use SNS (e.g., eHealth systems)? Although there is a hint of practical implication in line 238, it would be interesting to learn more. 

Author Response

We appreciate the opportunity to publish in your Journal and the comments received to improve our study report. Be aware that, following the suggestions of the reviewers, the title was changed from “Explaining the use of social network sites conceptualized as hedonic information systems in the elderly, Concepción, 2018” to “Explaining the use of social network sites as seen by older adults: the enjoyment component of a hedonic information system”.

Also the language was reviewed and corrected by a native English-speaking person. All the grammatical and spelling errors were corrected. The references were also reorganized, given new additions. An annex is located at the end of this document, with the results of additional analysis that were asked by the reviewer 2.

The following changes were made:

Comments Reviewer 1

Changes made

Title: I believe the title does not need to include   the location and the year of research setting. Hence, I may delete   ‘Concepcion, 2018’ from the title.

Place and year were removed from the title.

Additionally, the title was changed according to the   suggestion of the reviewer 2.

The new title is: “Explaining the use of social   network sites as seen by older adults: the enjoyment component of a hedonic   information system”.

Pg 1, line 30: It is unclear what the   authors mean by ‘limits of the personal life of its users’. Does it mean that   SNS contents were mostly limited to users’ personal lives? Please explain.

·  line 32: large corporations

·  line 36: What does ‘a single type   of social structure’ mean? Please explain the type that emerges among the SNS   users

Effectively, the interpretation of the reviewer is   correct. But the phrase was changes.

Capital letter was changed to lower case.

The sentence was reformed, hoping that it is better   understandable.

Pg 2, line 63-71: The sentences contain   grammatical mistakes.

a.       line 78: the adoption of SNS in older adults

b.        line 80-82: The sentence is grammatically problematic

c.     line   94-95: ‘That’ appears three times in one sentence. Need to be rephrased

All the sentences in Pg. 2, mentioned by the   reviewer, were reformed.

Pg 3, line 107-110: The sentence is too long and   unclear. Please rephrase.

a.       line   113: carried out from

b.      line   114: need period, instead of comma.

c.       Line   115: How do the authors extend the theoretical explanation in the current   research? Be specific.

d.      Line   115-120: Unclear sentences and hard to understand. Please rephrase.

e.       The   variable names need to be consistent across the manuscript. The variable   names in Figure 1 are completely different from those in the hypotheses. The   variable names in Table 1 and Table 3 are inconsistent as well.

f.       As   the authors suggest the mediating effects of perceived usefulness and   perceived enjoyment, the mediating relationships need to be hypothesized.

Pg. 3 (a-d) The sentences mentioned by the reviewer   were reformed.

e. The name of the variables was corrected and they   are consistent in all the text (Hypothesis), Tables and Figures.

f. The effect of mediation were analysed and there   is a paragraph describing the results at the end of the Results section.

We add an annex at the end of this document with the   results.

 Pg 5, Table 3: Please explain more about the   findings in Table 3. How should the composite reliability and AVE indicators   be interpreted? How can we determine the reliability and validity of data   using the indicators?

a.       Line   188: The authors show three R-squared values in Figure 2, and it is not clear   which one is related to which relationships. Please specify.

b.      line   191: Please explain the Bootstrapping method briefly and include citation.

Table 3 is explained completely, with all the   results in the correspondent section.

a. The figure was reorganized, incorporating the R2   inside the globes.

b. Bootstrapping method is described following the   sentences were it is mentioned

6.      Pg 6, line 200: Since the authors   suggest the mediating effect of perceived usefulness and perceived enjoyment,   these effects need to be hypothesized and tested accordingly. Please use   Sobel test or other appropriate methods to analyze the mediating effects.

a.       line 202: indicate that the use of

The mediation effect was   analysed and there is a paragraph describing the results at the end of the Results   section.

We add an annex at the end of this document with the   results.

a.The word was corrected.

Pg 7, line 229: The authors elaborated on the   theoretical implications of the research, but do not seem to mention   practical implications much. Given the findings, what should practitioners   consider when trying to encourage elders in Chile to use SNS (e.g., eHealth   systems)? Although there is a hint of practical implication in line 238, it   would be interesting to learn more

A paragraph describing more about the practical   implications in the health system was included at the end of the Discussion   section.

Reviewer 2 Report

I would like to thank the editor for giving me the opportunity to read this manuscript called Explaining the use of social network sites conceptualized as hedonic information systems in the elderly, Concepción, 2018.

It is research in a very important area about older adults use of Internet, especially social network sites, something that has come into focus as a tool for participation in society for this group in many different ways.

This particular research put focus on social networking for hedonic purposes as opposed to (or maybe in conjunction with) more utilitarian purposes.

The text is interesting, but it would be necessary to give more rigor and clarity in central aspects of the academic argumentation for this to be a publishable scientific text. The problematization leading up to the research aims, namely to determine the influence of perceptions of utility and enjoyment on the use of social network systems, is hard to follow and does not align with the title, result and the discussion. This is probably only a communicative aspect of the text, and I am sure the authors can clarify this. The two major issues I have with this and that must be clarified before it is possible to judge the whole text are the following:

1. If the purpose is (as it reads for this reviewer) to extend the TAM model with a Perceived Enjoyment factor to increase the explanatory power when looking at social network systems, how is this a conceptualization of a hedonic information system? From the problematization it reads as if the SNS could have both utilitarian and hedonic components which seems reasonable. It also does not align with the title. Make sure that the text is well aligned from title-problematization-result-discussion so that this is obvious and clear to a reader.

2. Please explain how and why the formulated hypotheses and the constructed model is the best way to go about this in the methods section? And as a consequence of this please elaborate on why a PLS analysis rather than the more traditional covariance-based or multi regression analysis gives the best fit.

When addressing these clarifications please also let a native English-speaking proofreader assist in improving the communicative aspects of the text. I suspect that some of the confusion might dwell in the partly unidiomatic language. This would also sort out the grammatical and spelling problems with the text.

On a separate note I would like to point out to the authors that the expression ‘the elderly’ has come to be considered ageist in many contexts. The phrases ‘older adults’ or ‘older persons’ are preferred choices.

Author Response

We appreciate the opportunity to publish in your Journal and the comments received to improve our study report. Be aware that, following the suggestions of the reviewers, the title was changed from “Explaining the use of social network sites conceptualized as hedonic information systems in the elderly, Concepción, 2018” to “Explaining the use of social network sites as seen by older adults: the enjoyment component of a hedonic information system”.

Below we detail the changes made, according to the suggestions of the reviewers.

Also the language was reviewed and corrected by a native English-speaking person. All the grammatical and spelling errors were corrected. The references were also reorganized, given new additions. An annex is located at the end of this document, with the results of additional analysis that were asked by the reviewer 2.

Many thanks and kind regards on behalf of the authors.

Following changes were made

Comments of reviewer 2

Changes made

1. If the purpose is (as it reads for this reviewer)   to extend the TAM model with a Perceived Enjoyment factor to increase the   explanatory power when looking at social network systems, how is this a   conceptualization of a hedonic information system? From the problematization   it reads as if the SNS could have a both utilitarian and hedonic component   which seems reasonable. It also does not align with the title. Make sure that   the text is well aligned from title-problematization-result-discussion so   that this is obvious and clear to a reader

A paragraph describing the objectives of the study   was added at the end of the Introduction.

A complete section (Literature review) was added in   to the manuscript, explaining better the methodological bases of the analyses   carried out.

We hope that this addition, plus other explanations   of the methods (already mentioned), are useful for the audience to better   understand the content of the manuscript.

Please explain how and   why the formulated hypotheses and the constructed model is the best way to go   about this in the methods section? And as a consequence of this please   elaborate on why a PLS analysis rather than the more traditional   covariance-based or multi regression analysis gives the best fit.

A complete section (Literature review) was added in   to the manuscript, explaining better the methodological bases of the analyses   carried out. The formulation of the hypotheses was improved following the   suggestions of the reviewer 1.

In the Materials and Methods section (following   Table 1), the reasons why PLS is used as a method of analysis are described   at length in the new paragraph.

When addressing these clarifications please also let   a native English-speaking proof-reader assist in improving the communicative   aspects of the text. I suspect that some of the confusion might dwell in the   partly unidiomatic language. This would also sort out the grammatical and   spelling problems with the text

The text was corrected by a native English-speaking assistant.

Grammatical and spelling problems were solved.

On a separate note I would like to point out to the   authors that the expression ‘the elderly’ has come to be considered ageist in   many contexts. The phrases ‘older adults’ or ‘older persons’ are preferred   choices

The word “elderly” was replaced by older adults, in   every part of the text were it was written.

Round 2

Reviewer 1 Report

The authors have done a nice job in revising the manuscript. The revised one reads smoother. I only have a couple of minor comments.

1.     Title: no need to put a period at the end.

2. Tables 1 and 3: Use à SNS Use

Reviewer 2 Report

Thank you very much for addressing most the questions and concerns I had in my first review and doing changes accordingly to your text.I have nothing further to add.